# ENS-10: A Dataset For Post-Processing Ensemble Weather Forecasts

**Saleh Ashkboos**
ETH Zürich
saleh.ashkboos@inf.ethz.ch

**Langwen Huang**
ETH Zürich
langwen.huang@inf.ethz.ch

**Nikoli Dryden**
ETH Zürich
ndryden@ethz.ch

**Tal Ben-Nun**
ETH Zürich
talbn@inf.ethz.ch

**Peter Dueben**
ECMWF
peter.dueben@ecmwf.int

**Lukas Gianinazzi**
ETH Zürich
glukas@ethz.ch

**Luca Kummer**
ETH Zürich
lkummer@student.ethz.ch

**Torsten Hoefler**
ETH Zürich
htor@inf.ethz.ch

## Abstract

Post-processing ensemble prediction systems can improve the reliability of weather forecasting, especially for extreme event prediction. In recent years, different machine learning models have been developed to improve the quality of weather post-processing. However, these models require a comprehensive dataset of weather simulations to produce high-accuracy results, which comes at a high computational cost to generate. This paper introduces the ENS-10 dataset,[1] consisting of ten ensemble members spanning 20 years (1998–2017). The ensemble members are generated by perturbing numerical weather simulations to capture the chaotic behavior of the Earth. To represent the three-dimensional state of the atmosphere, ENS-10 provides the most relevant atmospheric variables at 11 distinct pressure levels and the surface at $0.5°$ resolution for forecast lead times T=0, 24, and 48 hours (two data points per week). We propose the ENS-10 prediction correction task for improving the forecast quality at a 48-hour lead time through ensemble post-processing. We provide a set of baselines and compare their skill at correcting the predictions of three important atmospheric variables. Moreover, we measure the baselines' skill at improving predictions of extreme weather events using our dataset. The ENS-10 dataset is available under the Creative Commons Attribution 4.0 International (CC BY 4.0) license.

## 1 Introduction

Weather forecasting is among the most critical scientific applications, and has a significant influence on society and the economy [25]. Predicting the trajectory of a hurricane or tropical cyclone can save lives and billions of dollars in damages by allowing adequate mitigation efforts [29]. Moreover, weather forecasts can improve agricultural yield [53, 54] and guide decisions in aviation [16] and maritime ship transport [32]. However, predicting the weather, and extreme events in particular, are among the most complex tasks due to their chaotic behavior [35], and there has been an enormous amount of effort to tackle this problem in recent years [1, 4, 27, 48].

In the last decades, forecasting weather variables (like temperature or precipitation) has been done by solving complex partial differential equations (PDEs) [28, 30]. To this end, Numerical Weather

---

[1]Dataset and all its scripts can be found at https://github.com/spcl/ens10.

36th Conference on Neural Information Processing Systems (NeurIPS 2022) Track on Datasets and Benchmarks.

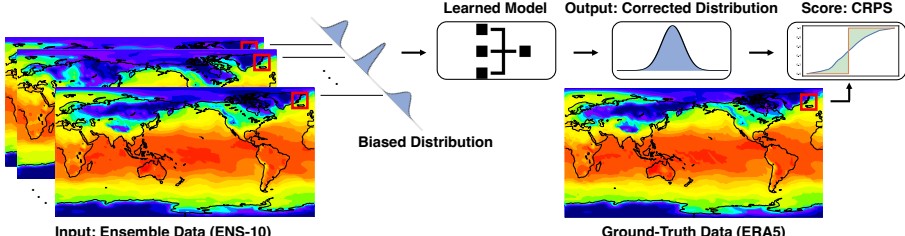

Figure 1: The post-processing pipeline feeds the ENS-10 ensemble, representing a biased distribution, to the deep learning model, which predicts a *corrected* distribution. This corrected distribution is scored against ERA5 ground truth data with the Continuous Ranked Probability Score (CRPS).

Prediction (NWP) models start from an initial weather condition and output the new conditions by numerically approximating a set of PDEs. As a single simulation cannot represent the chaotic behavior of the environment [35], *ensembles* of simulations have been introduced to quantify such uncertainty and improve forecast quality [6]. In such methods, the initial conditions of the environment are perturbed multiple times, and an NWP model is run on each condition to generate an ensemble of forecasts. However, running NWP models incurs high computational costs. In addition, ensemble weather prediction suffers from systematic errors, known as biases, which are introduced during either the generation of initial conditions or the simulation itself [47].

Ensemble post-processing approaches have been proposed to improve forecast skill by correcting the output distribution of the ensemble weather prediction models to remove biases, known as *prediction correction*, and are critical to improving the quality of ensemble forecasts [6, 45, 55]. In recent years, deep learning models [2, 12, 15, 26, 36, 39] have shown promising results on prediction correction over the well-known statistical methods [48] (e.g., EMOS [14], Bayesian Model Averaging [37]), which are representative of current approaches in production forecasting systems. Figure 1 summarizes the key steps of a deep learning-based post-processing pipeline. However, the performance of these models heavily depends on the available training data and the community lacks a unified and standard ensemble dataset, with different works using different forecast models, regions of the Earth, and so on [12, 26, 39]. Further, constructing an ensemble requires running multiple numerical models at a high computational cost. This makes training and comparing the quality of machine learning models for prediction correction challenging.

To address this challenge, we introduce the ENS-10 dataset, a collection of 10 perturbed ensemble members over 20 years (1998–2017) for medium-range ensemble forecasts (48 hour lead time). To represent the state of the atmosphere, we provide the most relevant atmospheric fields at 11 different pressure levels as well as on the surface on a structured longitude/latitude grid with $0.5°$ resolution. We then define the prediction correction task and propose two metrics of skill: Continuous Ranked Probability Score (CRPS), and the novel Extreme Event Weighted Continuous Ranked Probability Score (EECRPS). The weighted score gives importance to points where the ensemble members deviate from the climatology, rewarding skill on challenging, potentially unusual weather phenomena. We provide a set of baselines to apply unsupervised, statistical, and DNN-based supervised learning solutions (with different architectures) to the prediction correction task to serve as an initial benchmark. Finally, we provide a set of Python interfaces to download, train, and compare different models.

## 1.1 Related Work

Current weather forecasting is based on purely physical models, which numerically solve underlying equations to simulate weather [4]. In contrast, data-driven forecasting uses a learned model to directly predict weather variables. Such methods start from a deterministic atmospheric condition and generate the next state using a neural network or other model. In recent years, different architectures like fully connected networks [8], convolutional networks [8, 40, 43, 44], LSTMs [51], and transformers [34] have been applied to tackle this problem. While such models are becoming competitive for short-term nowcasting [10, 46], they currently are not competitive with NWP for medium-range forecasting [38].

As data-driven forecasting approaches heavily depend on the data, they usually use *reanalysis* data, namely reproducing historical weather data consistently with an existing NWP model, as ground truth. To this end, different datasets such as ERA5 [20] and CFSR [42] have been developed and

used as the input of the neural networks. Recently, WeatherBench [38] provided a set of benchmarks and metrics on a subset of the ERA5 dataset to accelerate data-driven weather forecasting research.

In contrast to deterministic data-driven forecasting, we focus on ensemble processing methods [48], which can easily slot into production forecast workflows [45]. These methods operate on a set of ensemble members, each generated by running a NWP simulation on perturbed initial atmospheric conditions. The goal is to correct the prediction by mitigating the biases of the ensemble outputs, known as ensemble post-processing. Statistical methods such as ensemble model output statistics (EMOS) [14] and Bayesian moving averages (BMA) [37] are widely used for correcting such biases.

Recent work has developed deep learning models for ensemble post-processing. To this end, different architectures, e.g., shallow fully connected networks [12, 39], LeNet-style convolutional networks [26], and U-Net-style networks [15] have been proposed. However, each work utilizes its own dataset and loss metrics (e.g., different regions of the Earth, different spatial or temporal resolutions, different forecast models, etc.), which makes comparisons challenging. ENS-10 fills this gap by providing a standard dataset and easy-to-use benchmarking infrastructure for post-processing.

Other weather datasets contain ensembles, but most are not suitable for our prediction correction task. For example, the ERA5 dataset [20] contains ensembles, but is not suitable for post-processing because it is a reanalysis, not a forecast; and the Large Ensemble Community Project [22] focuses on longer-term climate predictions. Most closely related to ENS-10 are the Global Ensemble Forecast System (GEFS) reforecast datasets [18, 31] and the THORPEX Interactive Grand Global Ensemble (TIGGE) [5]. However, new GEFS reforecasts use only five ensemble members, compared to ten in ENS-10. TIGGE consists of operational forecasts, as opposed to reforecasts, and therefore the forecast model, quality, and other parameters are inconsistent over time.

## 2 ENS-10 Dataset

### 2.1 Data Collection

Global weather predictions are performed operationally in certain time intervals and often use ensembles of simulations for uncertainty quantification — for example, a 50-member ensemble twice a day at the European Centre for Medium-range Weather Forecasts (ECMWF).

To derive a long-term training dataset that provides consistent quality for ensemble simulations, we use *reforecasts* [19], which are ensemble weather predictions that run routinely at ECMWF to simulate the weather over the past 20 years to the present. The resulting dataset is used to provide an estimate of the climate for each date to measure the *skill* of the forecast system [49, 50]. (Note the model used to generate reforecast datasets typically changes frequently as it is improved, so different reforecast datasets cannot typically be combined.) The reforecasts that were used for this dataset were generated by a model with 91 vertical levels using the Integrated Forecast System (IFS) of ECMWF [9], widely considered to be the most accurate global forecast model [17]. Reforecasts are generated for the time period from January 1998 to December 2017 and two simulations are started per week. The model uses a spectral representation of the model fields and a cubic octahedral reduced Gaussian grid is used to represent fields in grid-point space with a grid-spacing of approximately $36\,\mathrm{km}$. For all years, IFS model cycle Cy43r1 was used to generate the data for all dates before 5th June and model cycle Cy45r1 were used for later dates of each year. This configuration matches what is used for production forecasts at ECMWF. As machine learning solutions often make use of convolutions in horizontal directions, we map the model fields and the ERA5 dataset onto a structured longitude/latitude grid with a $0.5°$ resolution. The dataset is approximately 3 TB in size.

### 2.2 Dataset Features

The IFS weather model uses ten prognostic variables per grid point and 91 vertical levels per column, plus additional prognostic variables at the surface, yielding about one thousand prognostic variables per grid column. The vertical levels are processed into pressure levels, which are more representative for post-processing tasks. To keep data volume reasonable (the raw data would be hundreds of terabytes), we select a majority of the variables on each of a subset of the pressure levels, and additional surface variables. In doing so, we aim to select the input fields that hold the most significant amount of information about the three-dimensional state of the atmosphere.

Table 1: Parameters of ENS-10. The dataset contains each parameter for 10 different ensemble members on resolution 0.5° latitude/longitude grids at 0, 24, and 48 hour forecast lead times.

| Name | Abbr | Unit | 48 h forecast min, mean, max (1998–2015) |
|---|---|---|---|
| **Surface**: | | | |
| Sea surface temperature | SST | K | 309 / 286 / 281 |
| Total column water | TCW | $\mathrm{kg/m^2}$ | 133 / 25 / 0 |
| Total column water vapor | TCWV | $\mathrm{kg/m^2}$ | 94.3 / 24.4 / 0 |
| Convective precipitation | CP | m | 0.8552 / 0.0032 / 0 |
| Mean sea level pressure | MSL | Pa | 107485 / 101138 / 90724 |
| Total cloud cover | TCC | (0–1) | 1 / 0.62 / 0 |
| 10 m U wind component | U10m | m/s | 41.2 / -0.4 / -42.2 |
| 10 m V wind component | V10m | m/s | 46.2 / 0.2 / -42.9 |
| 2 m temperature | T2m | K | 324 / 287 / 191 |
| Total precipitation | TP | m | 1.16 / 0.01 / 0 |
| Skin temperature at surface | SKT | K | 334 / 288 / 189 |
| **Pressure levels** (10, 50, 100, 200, 300, 400, 500, 700, 850, 925 and 1000 hPa): | | | |
| U wind component | U | m/s | 150 / 6 / -116 |
| V wind component | V | m/s | 117 / 0 / -129 |
| Geopotential | Z | $\mathrm{m^2/s^2}$ | 316067 / 95656 / -7659 |
| Temperature | T | K | 325 / 248 / 165 |
| Specific humidity | Q | $\mathrm{kg\,kg^{-1}}$ | 0.0301 / 0.0026 / -0.000032 |
| Vertical velocity | W | Pa/s | 16.4 / 0 / -19.3 |
| Divergence | D | 1/s | 0.0014 / 0.0000 / -0.0017 |

ENS-10 consists of the atmospheric fields that appear most relevant in this context (see Table 1), selected using a combination of community consensus (e.g., [15, 34, 38, 39]) and expert meteorological advice from ECMWF. Some of the fields are available at the surface (e.g., the wind at 10 meter height in the zonal and meridional directions, U10m and V10m; or temperature at 2 meters above ground, T2m). Other fields are integrated quantities over the vertical column of the atmospheric model resulting in two-dimensional horizontal fields that are classified as surface fields (e.g., total column water, TCW). Finally, some of the fields are diagnosed at pressure levels along the vertical direction of the atmosphere. These are geopotential, which provides information about the larger synoptic situation; temperature and the U and V wind components, which are highly relevant for forecast users; and specific humidity, vertical velocity, and divergence, which provide information about the cloud fields and convective state of the grid column. ENS-10 provides these fields at 11 distinct pressure levels between 10 and 1000 hPa, to describe the state of the atmosphere from the surface to the stratosphere. The fields are at lead times of 0, 24, and 48 hours starting from two dates per week over a span of 20 years (1998–2017).

### 2.2.1 Surface Level Features

The following variables are provided at surface level[2].

**Sea Surface Temperature (SST)** is the temperature of the sea in units of kelvin.

**Total Column Water (TCW)** is the sum of water vapor, liquid water, cloud ice, rain, and snow in a column extending from the surface of the Earth to the top of the atmosphere. It is saved in units of kilogram per square meter.

**Total Column Water Vapor (TCWV)** is the amount of water vapor in a column extending from the surface of the Earth to the top of the atmosphere. Like TCW, it is saved in units of kilogram per square meter.

**Convective Precipitation (CP)** is the accumulated liquid and frozen water, comprising rain and snow, that falls to the Earth's surface, which is generated by the convection scheme in units of meters.

---

[2]Full definitions at https://apps.ecmwf.int/codes/grib/param-db/

**Mean Sea Level pressure (MSL)** is the pressure (force per unit area) of the atmosphere adjusted to the height of mean sea level in units of pascals.

**Total Cloud Cover (TCC)** is the proportion of a grid column covered by cloud calculated as a single level field from the clouds occurring at different model levels through the atmosphere.

**10m U wind component (U10m)** is the eastward component of the wind 10 meters above the ground in units of meters per second.

**10m V wind component (V10m)** is the northward component of the wind 10 meters above the ground in units of meters per second.

**2m Temperature (T2m)** is the temperature of air at 2 m above the surface of land, sea or in-land waters calculated by interpolating between the lowest model level and the Earth's surface, taking account of the atmospheric conditions. The field is saved in units of kelvin.

**Total Precipitation (TP)** is the accumulated liquid and frozen water, comprising rain and snow, that falls to the Earth's surface. It is the sum of large-scale precipitation and convective precipitation and saved in units of meters.

**Skin temperature at the surface (SKT)** indicates the temperature of the uppermost surface layer, which has no heat capacity and so can respond instantaneously to changes in surface fluxes, in kelvin.

### 2.2.2 Volumetric Features

The dataset provides the following seven fields at 11 distinct pressure levels between 10 and 1000 hPa, to describe the state of the atmosphere from the surface to the stratosphere.

**U wind component (U)** is the eastward component of the wind at each pressure level. It is the horizontal speed of air moving towards the east, in units of meters per second. A negative sign thus indicates air movement towards the west.

**V wind component (V)** is the northward component of the wind at each pressure level. It is the horizontal speed of air moving towards the north, in units of meters per second. A negative sign thus indicates air movement towards the south.

**Geopotential (Z)** is the gravitational potential energy of a unit mass, at a particular location, relative to mean sea level at each pressure level. It is also the amount of work that would have to be done, against the force of gravity, to lift a unit mass to that location from mean sea level. The variable is saved in units of $m^2/s^2$.

**Temperature (T)** shows the temperature at each pressure level. The field is saved in units of kelvin.

**Specific humidity (Q)** is the mass of water vapor per kilogram of moist air at each pressure level. The total mass of moist air is the sum of the dry air, water vapor, cloud liquid, cloud ice, rain and falling snow. The field is saved in units of kilograms per kilogram.

**Vertical Velocity (W)** is the air speed in the upward or downward direction at each pressure level in units of pascals per second.

**Divergence (D)** is the rate of spreading the air from a point per square meter at each pressure level in the inverse second. The negative values show that the air is concentrating.

## 3 Task

The ultimate goal of ensemble post-processing is to correct the forecast of an ensemble by adjusting for any biases present (see Figure 2). In this section, we define the prediction correction task for different variables using the ENS-10 dataset. We provide different evaluation metrics to assess the quality of a given solution for this task on weather and extreme event forecasting. We also study the role of varying the number of ensemble members on the quality of a solution in our task.

### 3.1 Prediction Correction

For a given time $T$, the input is a set of ensemble members $E = \{E_{k,T}\}_{k \in [1,10]}$. Each ensemble member $E_{k,T}$ contains all surface and volumetric features at time steps $T, T + 24\,\mathrm{h}$, and $T + 48\,\mathrm{h}$.

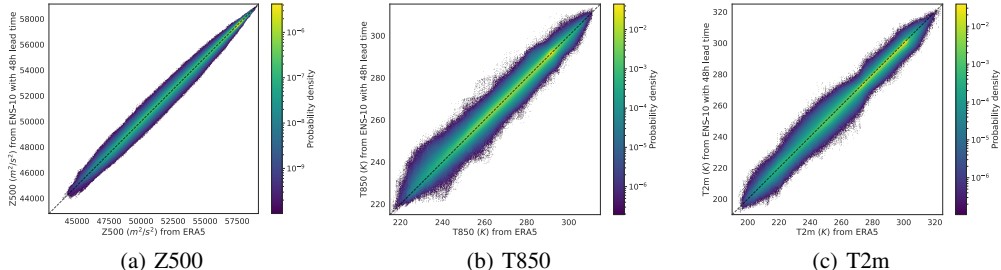

| (a) Z500 | (b) T850 | (c) T2m |

Figure 2: A scatterplot of the raw ENS-10 ensemble forecast against the ERA5 reanalysis reveals varying levels of uncertainty for different target variables. Moreover, certain ranges of values are better characterized by the raw forecast than others. Post-processing corrects the raw ensemble distribution to remove biases and improve the forecast skill.

For a given target variable at a specific pressure level (or at the surface), the task is to predict a *corrected* cumulative distribution function (CDF) $F^{i,j}$ for that variable at time $T + 48\,\text{h}$ at each grid point $(i, j)$ (at the specified pressure level). Usually, this is achieved by estimating the parameters of the output probability distribution. Many previous works [15, 36, 39] assume a Gaussian distribution at each grid point and estimate the means and standard deviations. We follow this approach for our baselines. However, in general, any distribution or distribution representation may be assumed.

In particular, we consider three target variables: **2 m temperature** (T2m), **temperature at 850 hPa** (T850), and **geopotential at 500 hPa** (Z500). The years 1998–2015 are used as the training set and 2016–2017 as the test set. We use ERA5 as ground truth data for the exact weather at prediction time.

### 3.2 Evaluation Metrics

To evaluate a potential solution for our task, we use two different metrics as scores for prediction skill. **CRPS** is used for scoring predictions in general, and is widely used for evaluating forecast skill (e.g., in [15, 17, 39]). We also define a novel metric, called **EECRPS**, which assesses the correction's improvement in skill for extreme event prediction.

**Continuous Ranked Probability Score (CRPS).** The corrected CDF $F$ is evaluated against the ERA5 ground-truth with CRPS [57]. CRPS generalizes the mean absolute error for the case where forecasts are probabilistic. Let $\mathbb{1}_{x \leq y}$ be an indicator function (1 if $x \leq y$ and 0 otherwise). Given the ground truth observation $x$ at grid-point $(i, j)$, the CRPS for the corrected CDF $F$ at point $(i, j)$ is

$$\text{CRPS}(F^{i,j}, x) = \int_{-\infty}^{\infty} \left( F^{i,j}(y) - \mathbb{1}_{x \leq y} \right)^2 dy, \tag{1}$$

where the integration is over the range of possible outcomes for the target variable (see Figure 3(a)). The CRPS assumes $F$ has finite expectation. We report the mean CRPS over all grid points over the two test years, and for each day, we consider the variable at 00:00 UTC. The CRPS integral can often be solved in closed form using an identity by Baringhaus and Franz [3, 13].

**Extreme Event Weighted Continuous Ranked Probability Score (EECRPS).** A crucial goal in correcting prediction biases is mitigating uncertainty during extreme weather events. In order to avoid muddling such events with forecast skill in the average case, we define a weighted version of CRPS that emphasizes such events.

A widely used metric that defines the irregularity of forecast field values is the extreme forecast index (EFI) [24, 58]. The EFI measures the deviation of the ensemble forecast relative to a probabilistic climate model (see Figure 3(b)). The EFI ranges between $-1$ and $1$, where large absolute values indicate a larger deviation from the meteorological record. Generally, values of magnitude between $0.5 - 0.8$ are considered *unusual* and values above $0.8$ are considered *very unusual* and indicate that extreme weather is likely. Therefore, given the ground-truth observation $y$ at grid-point $(i, j)$, we use the absolute value of EFI at that point to weight the CRPS as a secondary score:

$$\text{EECRPS}(F^{i,j}, y) := |\text{EFI}_{(i,j)}| \times \text{CRPS}(F^{i,j}, y). \tag{2}$$

We report the mean EECRPS over all grid points of the test years.

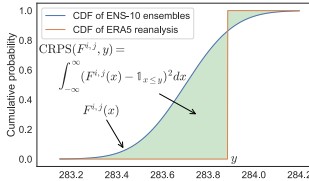
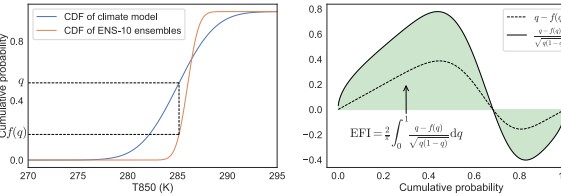

(a) CRPS is the calculated by integrating the square of the difference between the ensemble CDF and the reanalysis CDF (green area).

(b) **Left**: $f(q)$ is the cumulative probability of the ensemble values that is smaller than the $q$-quantile of the climate model. **Right**: EFI integrates the difference between the model climate and the ensemble forecast, but gives more weight to the extreme values.

Figure 3: Visualization of CRPS and EFI for predicting the temperature at $850\,\mathrm{hPa}$.

Given a grid-point $(i, j)$, for each date, the climate model is an empirical distribution function of all ensemble predictions in a window around that date over many years at $(i, j)$. That is, a sliding window is used to select the ensembles for each date.

We calculate the $\mathrm{EFI}_{(i,j)}$ for all grid-points $(i, j)$ over the ENS-10 test set with $0.05$-quantiles. To this end, at a specific date, we first calculate the cumulative histogram of the 10 ensemble members of our dataset at that date. Note that we use the same lead time of $48\,\mathrm{h}$ for all ensemble forecasts. Then, we aggregate ensemble predictions in a window of 9 dates centered around the selected date over the past 18 years. Hence, a total of $18 \times 9 \times 10$ ensemble members represent the climate model for each date. We calculate their quantiles from probability 0 to 1 at an interval of 0.05 as a discrete representation of the inverse of the cumulative distribution function. In the end, the $\mathrm{EFI}_{(i,j)}$ is calculated as

$$\mathrm{EFI}_{(i,j)} = \frac{2}{\pi} \sum_{q \in \mathcal{Q}} \frac{q - f_{(i,j)}(q)}{\sqrt{q(1-q)}} \Delta q, \tag{3}$$

where $\Delta q = 0.05$, $\mathcal{Q} = \{0, 0.05, 0.1, ..., 0.95, 1\}$, $f_{(i,j)}(q)$ is calculated from the $q$-quantile value of our climate model subtracted from the cumulative histogram at the position of the $q$-quantile value.

## 4 Benchmarks and Results

We now define several representative baselines for the prediction correction task on ENS-10, and evaluate them all with both CRPS and EECRPS using five and ten ensemble members as input. All baselines are implemented in PyTorch [33] and included in the ENS-10 repository.

### 4.1 Baseline Models

A handful of models have been developed to tackle the ensemble post-processing task during the past decades [12, 14, 15, 37, 39]. However, they fall into five broad categories from which we select exemplars: "Simple" learned statistical methods, such as EMOS [14, 15, 39], which are also broadly representative of what is presently used by production forecast systems for post-processing; MLPs [12, 39]; simple convolutional neural networks [26] (also widely used for weather prediction [38, 40, 52]); U-Nets [15]; and transformers, which have recently been applied to ensembles [11] (and also to weather prediction [34]). Our goal is to train these models to minimize CRPS. However, in general, CRPS is not a differentiable metric. To overcome this issue, following previous work [15, 36, 39], we assume a Gaussian distribution on the target variable and learn the mean and standard deviation of this distribution. In this case, the CRPS from Eqn. (1) can be rewritten as

$$\mathrm{CRPS}(F^{i,j}, x) = \sigma \left[ 2\psi\left(\frac{x-\mu}{\sigma}\right) + \frac{x-\mu}{\sigma}\left(2\phi\left(\frac{x-\mu}{\sigma}\right) - 1\right) - \frac{1}{\sqrt{\pi}} \right], \tag{4}$$

where $\mu$ and $\sigma$ are the mean and standard deviation of the distribution, and $\psi$ and $\phi$ are the probability density and cumulative density function of a standard Gaussian random variable, respectively [13].

In the simplest case, one can use the mean and standard deviation of the raw ensemble members as a predictor without any correction. However, to improve forecasting quality, the mean and standard deviation can be estimated using simple statistical approaches [14, 37] or deep learning models [12, 15, 39]. We define our baselines next.

The **raw** baseline refers to using the raw ensemble mean and standard deviation for prediction. We apply this baseline for predicting all three variables in the prediction correction task. To this end, we calculate the CRPS using the mean and standard deviation of the ensembles on each grid point at $T = 48\,\text{h}$ on our test set. We also calculate the EECRPS for T2m at $T + 48\,\text{h}$ on the test set.

**Ensemble Model Output Statistics (EMOS)** [14] is a statistical method based on a linear relation between the ensemble values and the corrected mean and variance.

For a given set of ensemble members $E = \{E_{i,T}\}_{i \in [1,m]}$, EMOS assumes a Gaussian distribution over the variable of interest $Y \sim \mathcal{N}(\mu, \sigma^2)$ on each point and computes the mean $\mu$ and standard deviation $\sigma$ using $\mu = a + \sum_{i=1}^{m} b_i E_{i,T}$ and $\sigma = c + d\sigma_E$, where $\sigma_E$ is the standard deviation of the ensemble variables in $E$. However, for predicting a target variable (such as Z500), EMOS uses only the input features for Z500 and does not exploit the inter-variable relations.

A **multi-layer perceptron (MLP)** is a series of fully-connected layers followed by a non-linear function. For our baseline, we use an MLP with one hidden layer, following Rasp and Lerch [39]. The network corrects the prediction point-wise by estimating the corrected mean and standard deviation of a corrected Gaussian random variable. For each point, we provide the mean and standard deviation of all variables in ENS-10 at lead time $T = 48\,\text{h}$ in the same pressure level as the variable of interest. In addition, we provide the spatial information of that point (latitude and longitude) in our baselines using a three-dimensional xyz coordinate on a unit sphere where $x = \cos(\psi)\sin(\phi), y = \cos(\psi)\cos(\phi), z = \sin(\psi)$ and $\psi, \phi$ are latitude and longitude.

The MLP outputs two values associated with the mean and the standard deviation of the variable of interest. As we normalize the input to the network, to get the correct mean and standard deviation of a given input, we multiply the first output value with the standard deviation of the ensemble member and add the ensemble mean at that point. Similarly, we get the corrected standard deviation by taking the exponential of the second output value and multiplying it with the ensemble standard deviation. The baseline network has 25 and 17 input dimensions (two inputs for each variable in the ENS-10 dataset and three for XYZ coordinates) for the surface and volumetric features, respectively. We use a hidden layer with dimension 128 and ReLU as the activation function.

Convolutional neural networks have demonstrated promise for both ensemble post-processing [26] and weather prediction [38, 40, 52]. We therefore follow Li et al. [26] and use a **LeNet**-style network as a baseline. Our network consists of three convolutional layers, each with an ELU [7] activation and batch normalization [21], kernel size $3 \times 3$, no padding, and 64, 32, and 16 channels, respectively. When post-processing a grid point, we input a $7 \times 7$ patch centered on that grid point. The output (of shape $16 \times 1 \times 1$) is then used as input to an MLP constructed as above, with the same three-dimensional location embedding concatenated. The output features of the network then encode the mean and standard deviation of the corrected grid point and are also handled as above.

The **U-Net** architecture is widely used in image segmentation tasks [41] and other tasks such as pixel-wise regression [56]. Recently, a U-Net style network has been applied to correcting the biases of ensembles [15]. We use a U-Net network as our baseline to predict the corrected mean and standard deviation of the Gaussian target variables.

Following Grönquist et al. [15], we create a U-Net baseline with three levels, each containing a set of [convolution, batch normalization, ReLU] modules. The network operates on the whole grid with 22 and 14 input dimensions (two inputs for each variable in the ENS-10 dataset) for the surface and volumetric features, respectively. We use convolutions with 32, 64, and 128 output channels in our network's first, second, and third levels, respectively.

Input data to the U-Net are normalized. The network outputs two grids for the mean and standard deviation of the variable of interest. Similar to the MLP, we scale and add the value of the first output channel with the standard deviation and mean of the ensemble members in each grid point. Also, we take the exponential of the second output channel of the network and multiply it with the ensemble standard deviation to get the corrected standard deviation.

Finally, following Finn [11], we apply a self-attentive ensemble **transformer** as a baseline. Our model consists of three main components: the embedding layer, the self-attentive transformer module, and the output layer. The embedding consists of three $5 \times 5$ convolutional layers with ReLU activations and eight filters. Data are circularly padded longitudinally and zero padded latitudinally. Then, for the $i$-th ensemble member, the self-attentive transformer module uses the weights estimated

Table 2: Global mean CRPS and EECRPS on the ENS-10 test set (2016–2017) for baseline models with five (5-ENS) or ten (10-ENS) ensemble members.

| Metric | Model | Z500 $[\mathrm{m}^2\,\mathrm{s}^{-2}]$ | | T850 [K] | | T2m [K] | |
|---|---|---|---|---|---|---|---|
| | | 5-ENS | 10-ENS | 5-ENS | 10-ENS | 5-ENS | 10-ENS |
| CRPS | Raw | 81.03 | 78.24 | 0.748 | 0.719 | 0.758 | 0.733 |
| | EMOS | $79.08^{\pm0.739}$ | $81.74^{\pm6.131}$ | $0.725^{\pm0.002}$ | $0.756^{\pm0.052}$ | $0.718^{\pm0.003}$ | $0.749^{\pm0.054}$ |
| | MLP | $75.84^{\pm0.016}$ | $74.63^{\pm0.029}$ | $0.701^{\pm2e-4}$ | $0.684^{\pm4e-4}$ | $0.684^{\pm6e-4}$ | $0.672^{\pm5e-4}$ |
| | LeNet | $\mathbf{75.56}^{\pm0.101}$ | $\mathbf{74.41}^{\pm0.109}$ | $0.689^{\pm2e-4}$ | $0.674^{\pm2e-4}$ | $0.669^{\pm7e-4}$ | $0.659^{\pm4e-4}$ |
| | U-Net | $76.66^{\pm0.470}$ | $76.25^{\pm0.106}$ | $0.687^{\pm0.003}$ | $0.669^{\pm0.009}$ | $0.659^{\pm0.005}$ | $0.644^{\pm0.006}$ |
| | Transformer | $77.30^{\pm0.061}$ | $74.79^{\pm0.118}$ | $\mathbf{0.686}^{\pm0.002}$ | $\mathbf{0.665}^{\pm0.002}$ | $\mathbf{0.649}^{\pm0.004}$ | $\mathbf{0.626}^{\pm0.004}$ |
| EECRPS | Raw | 29.8 | 28.78 | 0.256 | 0.246 | 0.258 | 0.25 |
| | EMOS | $29.10^{\pm0.187}$ | $30.13^{\pm2.166}$ | $0.248^{\pm3e-4}$ | $0.259^{\pm0.018}$ | $0.245^{\pm0.001}$ | $0.255^{\pm0.018}$ |
| | MLP | $27.86^{\pm0.006}$ | $27.41^{\pm0.010}$ | $0.240^{\pm1e-4}$ | $0.234^{\pm2e-4}$ | $0.233^{\pm2e-4}$ | $0.229^{\pm2e-4}$ |
| | LeNet | $\mathbf{27.72}^{\pm0.039}$ | $\mathbf{27.30}^{\pm0.037}$ | $0.235^{\pm5e-5}$ | $0.230^{\pm8e-5}$ | $0.228^{\pm2e-4}$ | $0.224^{\pm1e-4}$ |
| | U-Net | $27.98^{\pm0.240}$ | $27.61^{\pm0.490}$ | $0.235^{\pm0.003}$ | $0.230^{\pm0.002}$ | $0.223^{\pm5e-4}$ | $0.219^{\pm0.001}$ |
| | Transformer | $28.35^{\pm0.026}$ | $27.42^{\pm0.047}$ | $\mathbf{0.235}^{\pm0.001}$ | $\mathbf{0.227}^{\pm0.001}$ | $\mathbf{0.222}^{\pm0.001}$ | $\mathbf{0.214}^{\pm0.001}$ |

based on the query-key similarity, where the query represents the $i$-th ensemble member and the key the perturbations of the ensemble members. Finally, the output layer will map the transformed ensemble member with a $1 \times 1$ convolutional layer to the corrected ensemble member. We use one self-attentive transformer module with 16 heads. As the model outputs the corrected ensemble member, we calculate the mean and standard deviations offline to compute the CRPS and EECRPS.

## 4.2 Baseline Scores

We train all models using the Adam optimizer [23] with learning rate $10^{-5}$, except the transformer, which uses 0.001. We do not use any learning rate scheduler during training. Input data is split into mini-batches according to the time index: each sample inside a mini-batch contains a "slice" of data with the same time index but different latitude/longitude indices. We use a batch size of 8 for the U-Net model, and 1 for the EMOS, MLP, and transformer models, as larger batch sizes yielded degraded performance. For the LeNet model, we use batch size 1 but feed the entire grid into the network at once. We train all models for 10 epochs on a single A100 GPU. Training the models took 0.75 (EMOS), 0.25 (MLP), 1.25 (LeNet), 1 (U-Net), and 1 (transformer) hours, respectively.

The input data normalization for the MLP and U-Net models is performed according to the mean and standard deviation over all the time steps and ensemble members. Specifically, for each input variable at each latitude/longitude, the value is subtracted by the time-ensemble mean and divided by the standard deviation. In this way, any Gaussian distributed value over time and ensembles can be scaled to the Gaussian distribution with mean 0 and standard deviation 1. The normalization process is performed independently for different latitudes or longitudes to flatten the variance over different locations, following Grönquist et al. [15].

Following the original implementation of EMOS [14], the normalization of input data for the EMOS model linearly scales each variable to the range between 0 and 1 using minimum and maximum values over all the time steps, ensemble members, and grid points.

In addition to using the whole dataset (ten ensemble members), to show the importance of the number of ensemble members on the prediction correction task, we repeat all our experiments with the first five ensemble members (half of the dataset) from ENS-10. To evaluate the ability of our baselines on extreme event forecasting, we calculate the EECRPS of the T2m variable using the extracted EFI from our dataset (see Section 3.2 for more details). We use the absolute value of EFI to weight the CRPS metric before averaging over the global data points in our test set. We train each model using three different random seeds and report the mean and standard deviation.

We show the results of our baselines in Table 2. Performance trends are similar between both CRPS and EECRPS: the LeNet-style model is best on Z500, whereas the transformer is best on T850 and T2m. The U-Net model also performs consistently well across all metrics. EMOS exhibited degradation in performance relative to the raw ensemble when using ten ensemble members in some runs and consequently has a large spread in performance; we suspect this could be avoided with

additional optimizer tuning. Overall, no one model is consistently best, indicating the need to study a broad range of architectures and to refine their performance for future post-processing applications.

## 5 Conclusion

We present a dataset, ENS-10, and metrics for learning to correct biases in ensemble weather prediction systems. In particular, ten high-resolution ensemble members were generated for 20 years of weather forecasts. To represent the atmospheric environment of the earth, the dataset consists of the 11 most relevant surface variables as well as 7 variables at 11 distinct pressure levels in $0.5°$ resolution. To measure prediction correction quality, we benchmark learned models using a widely-used score (CRPS) and a novel scoring function that prioritizes extreme weather events (EECRPS). As a baseline, we evaluate a set of statistical and machine learning models that are used by the weather and climate community on the dataset. By generating a high-resolution dataset of perturbed ensemble member simulations, ENS-10 enables data-driven global scale uncertainty quantification and bias correction, aiding in accurate pinpointing of extreme weather events. However, we caution that ENS-10 is not suitable on its own for tasks such as weather *prediction*; and further that its performance evaluation is necessarily predicated on the quality of the ERA5 reanalysis dataset we use as ground truth.

Beyond ensemble post-processing, ENS-10 could be used for additional tasks, such as uncertainty quantification for weather prediction, developing probabilistic forecasts from deterministic trajectories, learning diagnostic fields from the prognostic fields of a forecast, or learning to fill gaps in weather time-series data to reduce the size of NWP model outputs.

The ENS-10 dataset can be downloaded directly[3] or through a downloader Python script. Examining the raw data, preprocessing, normalization, and training are all provided through a set of Python APIs[4], examples written in PyTorch, and Jupyter notebooks[5].

## Acknowledgements

This project received EuroHPC-JU funding under grant MAELSTROM, No. 955513. N.D. is supported by the ETH Postdoctoral Fellowship. T.B.N. is supported by the Swiss National Science Foundation (Ambizione Project #185778). We thank the Swiss National Supercomputing Center (CSCS) for providing computing resources.

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
