# OpenReview forum: "ENS-10: A Dataset For Post-Processing Ensemble Weather Forecasts"
_NeurIPS.cc/2022/Track/Datasets_and_Benchmarks — NeurIPS 2022 Datasets and Benchmarks _

### Official Review · Reviewer_Dzz8 · 2022-07-27
**A dataset for validation of weather prediction**

**Rating:** 5
**Confidence:** 5
**Correctness:** The experiments are appropriate and t…
**Clarity:** The paper is clearly written.

**Strengths:**

-- The paper is overall well written
-- The topic is very important and is not widely explored by the ML community.

**Weaknesses:**

There is nothing wrong with the paper but nothing stands out either. The data set is quite standard and there are multiple other sources to get a similar dataset. The metrics and methods are also conventional. It is not clear/not discussed what the ML community can do with this data beyond forecasting and bias correction.

**Additional Feedback:**

Do authors plan to regularly update these data? If yes, how will the maintenance be organized?

**Documentation:**

Satisfactory

**Relation To Prior Work:**

No proper discussion of competing data sets.

**Summary And Contributions:**

The paper discusses a dataset and metrics for mesoscale weather forecasting bias correction. The topic is undoubtedly very important and has broader societal implications in general.

---

> ### Author Response · Authors · 2022-08-19
> **Initial Response**
>
> > The data set is quite standard and there are multiple other sources to get a similar dataset. The metrics and methods are also conventional. It is not clear/not discussed what the ML community can do with this data beyond forecasting and bias correction.
>
> We respectfully disagree with this characterization. While there are many datasets for weather and climate prediction that are available (including for nowcasting, medium-range forecasting, and subseasonal / seasonal climate forecasting), there are far fewer for ensemble post-processing. Many datasets that do contain ensembles are not suitable (e.g., the ERA5 uncertainty ensembles cannot be used for post-processing because the dataset is a reanalysis, not a forecast; the NCAR Large Ensemble Community Project is focused on longer-term climate predictions rather than medium-range forecasting). Further, even among existing works on ensemble post-processing, different papers use different datasets (e.g., different forecast models, different spatial or temporal resolutions, different meteorological fields, etc.) which makes it difficult to compare results. For example:
> * Rasp et al. [1] use the TIGGE dataset with a different selection of meteorological fields (e.g., excluding temperature on pressure levels) only over Germany.
> * Ghaznavian et al. [2] uses precipitation data from the Global Ensemble Forecast System at 1 degree spatial resolution over California.
> * Li et al. [3] uses the ENS-10 dataset, but only over China.
>
> Further, the TIGGE dataset consists of operational ensemble forecasts, as opposed to reforecasts, and therefore the configuration, quality, etc. of the forecast models change over time. GEFS reforecasts (the latest version of which cover 2000-2019) consist of only five ensemble members.
>
> The community lacks a consistent and standardized dataset and benchmark, which ENS-10 aims to provide. If there is additional related work you think we should discuss, please let us know and we will be happy to incorporate it.
>
> While the use of CRPS to evaluate the prediction skill is standard, we do not view this as a downside, but rather it is exactly why we selected it, since it is widely used in the community and a critical metric for evaluating forecast models. Further, we also define the Extreme Event Weighted CRPS, which is, to our knowledge, novel. Similarly, we selected our baseline models on the basis of them being representative of existing methods (please see our general comment for more details).
>
> We have expanded our discussion of related work to better place ENS-10 in context. We have also added discussion of other tasks ENS-10 is suitable for, which include uncertainty quantification for weather prediction, developing probabilistic forecasts from deterministic trajectories, and learned compression for weather data.
>
> > Do authors plan to regularly update these data? If yes, how will the maintenance be organized?
>
> We do not plan to update the data (please see Section A.7 on maintenance). Because the production Integrated Forecast System which produces the hindcasts changes over time, the meteorological fields produced would change. Hence, data produced from different hindcasts is not reasonable for use in prediction correction.
>
> **References:**
>
> [1] Rasp & Lerch, “Neural Networks for Post-Processing Ensemble Weather Forecasts”, _Monthly Weather Review_ 146(11), 2018.
>
> [2] Ghazvinian et al., “A Novel Hybrid Artificial Neural Network - Parametric Scheme for Postprocessing Medium-range Precipitation Forecasts”, _Advances in Water Resources_ 151, 2021.
>
> [3] Li et al., “Convolutional Neural Network-based Statistical Post-Processing of Ensemble Precipitation Forecasts”, _Journal of Hydrology_ 605, 2022.

---

> ### Comment · Reviewer_Dzz8 · 2022-08-25
> **Thank you for the response**
>
> I appreciate the response provided by the authors. Still, I think that the set of the considered ML models is very limited and the considered CRPS is based on the Gaussianity assumption which is questionable for these data.  However, I'm fine to raise my score a bit.

---

> > ### Author Response · Authors · 2022-08-26
> > **Additional Response**
> >
> > Thank you for considering our response and updating your score.
> >
> > > I think that the set of the considered ML models is very limited
> >
> > With our updated paper, we now consider five baseline models in total (EMOS, an MLP, a CNN, a U-Net, and a transformer). To our knowledge, these broadly cover all classes of models considered for ensemble post-processing in prior work. If there are additional models you think we should consider, please let us know.
> >
> > > the considered CRPS is based on the Gaussianity assumption which is questionable for these data
> >
> > We want to clarify that CRPS assumes no underlying distribution. However, we follow prior work [1, 2, 3] in assuming a Gaussian distribution in order to have a closed-form representation for backpropagation. We additionally note that we explicitly wish to allow others using ENS-10 to assume different distributions in order to advance the state-of-the-art (see Section 3.1, line 199).
> >
> > **References:**
> >
> > [1] Grönquist et al., “Deep Learning for Post-Processing Ensemble Weather Forecasts”, Philosophical Transactions of the Royal Society A, 2021.
> >
> > [2] Rasp & Lerch, “Neural Networks for Post-Processing Ensemble Weather Forecasts”, Monthly Weather Review 146(11), 2018.
> >
> > [3] Peng et al., “Prediction Skill of Extended Range 2-m Maximum Air Temperature Probabilistic Forecasts Using Machine Learning Post-processing Methods”, _Atmosphere_ 11(8), 2020.

---

> > > ### Comment · Reviewer_Dzz8 · 2022-08-26
> > > **ML models and Gaussianity**
> > >
> > > Thank you for the response. On the ML models, it looks as you're using a vanilla CNN (or quite close to it), it'd be better, of course, to consider more advanced SOTAs. On Gaussianity, how exactly do you compute CRPS? There are two ways: when CDF F is known and one based on the empirical estimates of moments. It is not clear which version you use as you have only a generic formula (1).

---

> > > > ### Author Response · Authors · 2022-08-27
> > > > **Answers/clarifications**
> > > >
> > > > Thanks for the quick response.
> > > >
> > > > > On the ML models, it looks as you're using a vanilla CNN (or quite close to it), it'd be better, of course, to consider more advanced SOTAs.
> > > >
> > > > We want to point out that, while the CNN baseline is relatively simple, we also evaluate on U-Nets (a more advanced convolutional baseline) and a self-attentive ensemble transformer (which incorporates convolution as well as attention). For users who want to try additional architectures, it is easy to add them to our benchmarking infrastructure.
> > > >
> > > > > On Gaussianity, how exactly do you compute CRPS? There are two ways: when CDF F is known and one based on the empirical estimates of moments. It is not clear which version you use as you have only a generic formula (1).
> > > >
> > > > We give the closed-form version of CRPS that we use in Equation 4 (Section 4.1), which is derived following [1]. This formulation was used by the prior work we mentioned before.
> > > >
> > > > **References:**
> > > >
> > > > [1] Gneiting et al., "Calibrated Probabilistic Forecasting Using Ensemble Model Output Statistics and Minimum CRPS Estimation", _Monthly Weather Review_ 133(5), 2005.

---

> > > > > ### Comment · Reviewer_Dzz8 · 2022-08-27
> > > > > **Gaussiniaty**
> > > > >
> > > > > If you use equation 4, it means that you assume Gaussianity (see Zamo and Naveau  (2018) "Estimation of the Continuous Ranked Probability Score with Limited Information and Applications to Ensemble Weather Forecasts" Mathematical Geosciences, 50, pp. 209–234 on CRPS forms with/without the assumption on the known CDF form and CRPS in case CDF is unknown). It is not a big deal really unless you have heavy tails and outliers, but it would also mean that showing other metrics besides CRPS would be beneficial.
> > > > >
> > > > > On more advanced CNNs, I understand it's the Dataset track but still the state-of-the-art models will be much more preferable than the basic ones, unless there is a good reason to show basic models. Note that U-Net is dated to 2015-2018 which cannot be viewed as SOTA either.

---

> > > > > > ### Author Response · Authors · 2022-08-28
> > > > > > **On SOTA Models**
> > > > > >
> > > > > > > On more advanced CNNs, I understand it's the Dataset track but still the state-of-the-art models will be much more preferable than the basic ones, unless there is a good reason to show basic models. Note that U-Net is dated to 2015-2018 which cannot be viewed as SOTA either.
> > > > > >
> > > > > > We believe U-Nets are either state-of-the-art or near-SOTA on many tasks in weather, and serve as a suitable baseline for ENS-10, especially when combined with our other baselines, particularly the self-attentive ensemble transformer, a more recent transformer baseline. In recent results in different weather forecasting benchmarks and competitions, U-Nets have been highly competitive:
> > > > > > * At the [2021 Weather4cast competition](https://www.iarai.ac.at/weather4cast/2021-competition/), two of the top three models were U-Net-style architectures [1].
> > > > > > * In the corresponding [2022 competition](https://www.iarai.ac.at/weather4cast/) at NeurIPS 2022, their provided baseline is [also a U-Net](https://github.com/iarai/weather4cast-2022#training).
> > > > > > * The second-best performing model (and the best iterative model) on the [WeatherBench](https://github.com/pangeo-data/WeatherBench) data-driven weather forecasting benchmark is a U-Net-style architecture [2].
> > > > > >
> > > > > > Beyond weather, vision transformer and U-Net variants seem to be consistently at or near SOTA for [semantic segmentation](https://paperswithcode.com/task/semantic-segmentation) and [medical image segmentation](https://paperswithcode.com/task/medical-image-segmentation).
> > > > > >
> > > > > > However, please keep in mind that our aim here is not to "solve" ENS-10 but rather to provide a set of baseline models that both represent prior work in ensemble post-processing and provide a starting point for the community to develop improved models. If there is a specific architecture you think would improve the benchmarks, we are happy to add it to the final version of the paper (please keep in mind the discussion/update period ends on August 29th).
> > > > > >
> > > > > > **References:**
> > > > > >
> > > > > > [1] Pedro et al., "High-Resolution Multi-Channel Weather Forecasting — First Insights on Transfer Learning from the Weather4cast Competitions 2021", IEEE Big Data 2021.
> > > > > >
> > > > > > [2] Weyn et al., "Improving Data-Driven Global Weather Prediction Using Deep Convolutional Neural Networks on a Cubed Sphere", _Journal of Advances in Modeling Earth Systems_ 12(9), 2020.

---

### Official Review · Reviewer_theC · 2022-07-27
**A dataset of ensemble weather forecast correction by using downstream Machine Learning methods.**

**Rating:** 7
**Confidence:** 3
**Correctness:** No incorrectness issues were found.
**Clarity:** Well-written article.

**Strengths:**

* The authors have provided a useful open source dataset (ENS-10) for ensemble weather prediction correction for 20 years along with a well-maintained website.
* A new error metric (EECRPS) has been introduced to evaluate predictions under extreme weather events.
* The dataset has been used in a predictive task of prediction correction using several Machine Learning models.

**Weaknesses:**

* As the authors discuss on page 7 of the manuscript, there exist many models in the literature for ensemble post-processing of weather forecasts. Nevertheless, the authors have attempted only a few of them (particularly only 2 Machine Learning methods in their predictive task), which seems a limited benchmarking attempt in my opinion.
* The benchmarking study focuses only on ensemble correction using different techniques. However, to make a fair comparison, other direct data-driven weather forecasting methods could also have been tried out (as the authors discuss on page 2), to establish the importance of performing ensemble correction and thus the usage of this dataset in the first place.
* The introduced dataset is 20 years long but only until 2017. How is it extended beyond 2017 for more recent weather forecasting work?

**Additional Feedback:**

The authors mention that 'ENS-10 consists of the atmospheric fields that appear most relevant in this context'. However, it is not clear or mentioned, on what basis the relevance of these atmospheric fields is evaluated.

**Documentation:**

Well-written documentation is provided along with the supplementary material and the website.

**Ethics:**

No ethical issues.

**Relation To Prior Work:**

Prior work is sufficiently discussed.

**Summary And Contributions:**

This article introduces and makes available an open-source weather prediction dataset spanning 20 years. The dataset can be readily utilized for ensemble prediction correction by other Machine Learning models for weather forecasting tasks. This work provides support for many weather forecasting tasks being carried out in other domains. The importance of ensemble weather correction has been demonstrated by a predictive task using the introduced dataset.

---

> ### Author Response · Authors · 2022-08-19
> **Initial Response**
>
> > As the authors discuss on page 7 of the manuscript, there exist many models in the literature for ensemble post-processing of weather forecasts. Nevertheless, the authors have attempted only a few of them (particularly only 2 Machine Learning methods in their predictive task), which seems a limited benchmarking attempt in my opinion.
>
> Please see our general comment, where we have added additional baselines and discussion. In brief, we selected baselines that were representative of prior work on ensemble post-processing and cover the key approaches used. Additionally, EMOS is also a learned method, although it does not use a neural network.
>
> > The benchmarking study focuses only on ensemble correction using different techniques. However, to make a fair comparison, other direct data-driven weather forecasting methods could also have been trieacd out (as the authors discuss on page 2), to establish the importance of performing ensemble correction and thus the usage of this dataset in the first place.
>
> We believe that the need for ensemble correction is well-established by the existing literature, and have added a brief discussion of this to the paper. Ensemble methods for forecasting have been in use for 20 - 30 years, and ensemble post-processing has been critical for ensuring forecast quality [1, 2, 3]. However, we note that post-processing in production systems is typically done with relatively simple methods (e.g., similar to EMOS) and using neural networks for this is a recent topic of interest.
>
> Further, while there is significant interest in exploring ML approaches for directly predicting weather, these currently provide worse results than conventional numerical weather prediction systems [4]. Additionally, the most recent approaches also rely on ensembles (e.g., [5]).
>
> > The introduced dataset is 20 years long but only until 2017. How is it extended beyond 2017 for more recent weather forecasting work?
>
> Please see our general comment.
>
> > The authors mention that 'ENS-10 consists of the atmospheric fields that appear most relevant in this context'. However, it is not clear or mentioned, on what basis the relevance of these atmospheric fields is evaluated.
>
> Please see our general comment. In short, to keep the dataset size reasonable, we selected key parameters based on community consensus and meteorological expertise.
>
> **References:**
>
> [1] Buizza & Richardson, “25 Years of Ensemble Forecasting at ECMWF”, _ECMWF Newsletter_ 153 (25), 2017. [http://dx.doi.org/10.21957/bv418o ]
>
> [2] Schultz et al., “Can Deep Learning Beat Numerical Weather Prediction?”, _Philosophical Transactions of the Royal Society A_ 379(2194), 2021.
>
> [3] World Meteorological Organization, “Guidelines on Ensemble Prediction System Postprocessing, 2021 Edition”, WMO-No. 1254, 2021.
>
> [4] Rasp et al., “WeatherBench: A Benchmark Data Set for Data-Driven Weather Forecasting”, _Journal of Advances in Modeling Earth Systems_ 12(11), 2020.
>
> [5] Pathak et al., “FourCastNet: A Global Data-driven High-resolution Weather Model using Adaptive Fourier Neural Operators”, arXiv 2022.

---

> > ### Comment · Reviewer_theC · 2022-08-26
> > **Thanks for the response**
> >
> > I would like to thank the authors for the detailed response to my comments. In light of these explanations that now give me a clearer picture of the paper, I would like to change my score to 7.

---

### Official Review · Reviewer_mdzJ · 2022-07-27
**Important dataset for weather prediction**

**Rating:** 6
**Confidence:** 4
**Correctness:** The claims look correct.
**Clarity:** The paper is well-written overall.

**Strengths:**

Overall, a strong dataset paper. The lack of large scale datasets is a major gap for the community working on data-driven weather forecasting, so this is an important paper.

**Weaknesses:**

1. From the perspective of releasing open-source datasets, wouldn't it be better to include all 10 prognostic variables per grid point? The reduction of features can be done by the users of the dataset.
2. I am a bit confused about using ERA5 as ground truth data for the exact weather at prediction time. What exactly are the labels in the data?
3. The authors write in the supplementary material that there are no sources of errors or noise in the dataset. Does the original data not suffer from noisy measurements?  Also, the spatial discretization must include some noise, right? The authors should clarify this.
4. Other than the e Prediction Correction task, what other tasks can the dataset be used for?
5. What guided the choice of the baselines?

**Additional Feedback:**

NA

**Documentation:**

The documentation (supplementary material) and the GitHub page are well-written and maintained.

**Relation To Prior Work:**

Yes.

**Summary And Contributions:**

I think the dataset serves an important purpose. Part of the reason why "physics-based" models have been more popular in weather prediction is the lack of large and principled datasets to train data-driven models. This dataset alleviates this challenge to some extent and is an important contribution. I am broadly in support of accepting the paper.  That being said, I have some questions about the dataset. My rating is conditional on the questions being answered.

---

> ### Author Response · Authors · 2022-08-19
> **Initial Response**
>
> > From the perspective of releasing open-source datasets, wouldn't it be better to include all 10 prognostic variables per grid point? The reduction of features can be done by the users of the dataset.
>
> Please see our general comment. In short, to keep the dataset size reasonable, we selected key parameters based on community consensus and meteorological expertise.
>
> > I am a bit confused about using ERA5 as ground truth data for the exact weather at prediction time. What exactly are the labels in the data?
>
> ERA5 is a reanalysis dataset providing best-estimate values for weather variables over time. Hence, it contains the actual values of the quantities ENS-10 ensemble members predict (e.g., what the 2 meter temperature is). This is used to evaluate the corrected CDF post-processing models produce on ENS-10.
>
> > The authors write in the supplementary material that there are no sources of errors or noise in the dataset. Does the original data not suffer from noisy measurements? Also, the spatial discretization must include some noise, right? The authors should clarify this.
>
> This is a good point, thank you. We have added a discussion of sources of uncertainty and error in the underlying data, including aleatoric, model, and structural uncertainties from observations, the numerical model, and other sources. We note, however, that the goal of the prediction correction task we use ENS-10 for is not to directly operate on meteorological measurements, but to correct the output biases of a numerical weather prediction model.
>
> > Other than the e Prediction Correction task, what other tasks can the dataset be used for?
>
> There are a number of other tasks one could use ENS-10 for, and we have added some discussion of this to the paper. In brief, you could use it for uncertainty quantification for weather prediction (e.g., Bayesian approaches combined with the WeatherBench forecasting benchmark); developing probabilistic forecasts from deterministic trajectories; learning diagnostic fields from the prognostic fields of a forecast; or to learn to fill gaps in weather time-series data to reduce the size of the model output (sometimes called “tethering”).
>
> > What guided the choice of the baselines?
>
> Please see our general comment, where we have added additional baselines and discussion. In brief, we selected baselines that were representative of prior work on ensemble post-processing.

---

### Official Review · Reviewer_u77G · 2022-07-28

**Rating:** 7
**Confidence:** 3

**Strengths:**

- Explains the challenge when predicting weather and the necessity to use ensemble processing methods instead of deterministic data-driven forecasting.
- The authors use reforecasts that run routinely at ECMWF to simulate weather over the past 20 years; this way, the resulting dataset can serve as a measure to skill a forecast system.

**Weaknesses:**

- Weather prediction is challenging because of its multiple variables, which makes it difficult to evaluate its usefulness or advantages.

**Additional Feedback:**

Do the authors consider other models for baseline models?

**Clarity:**

The paper is well written, very technical in terms of weather but easy to follow.

**Correctness:**

The paper is coherent when explaining data collection and metrics for its evaluation.

**Documentation:**

code available on Github.

**Ethics:**

There are no ethical concerns, but essential to consider that the authors are not claiming to use this dataset to predict the weather directly.

**Relation To Prior Work:**

Using ensemble processing methods instead of deterministic data-driven forecasting.

**Summary And Contributions:**

- The authors present a post-processing ensemble prediction system to improve weather forecasting.
- The task is to correct the prediction by post-processing the ensemble members.
- The metrics used in the paper are: the Continuous Ranked Probability Score (CRPS) and Extreme Event Weighted Continuous Ranked Probability Score (EECRPS). The authors proposed the last one, and its work is to assess the correction improvement in skill for extreme event prediction.
- In case of baseline methods: Raw, Ensemble Model Output Statistics (EMOS), Multi-layer perception(MLP), and U-Net.

---

> ### Author Response · Authors · 2022-08-19
> **Initial Response**
>
> > Weather prediction is challenging because of its multiple variables, which makes it difficult to evaluate its usefulness or advantages.
>
> Weather prediction is indeed a very challenging task, and how to trade off performance among different quantities is not always clear. Nevertheless, we believe our evaluation metrics are reasonable and in line with broader community choices [1, 2, 3, 4]. Additionally, we want to clarify that we are focused on ensemble post-processing with ENS-10, not directly predicting weather, and the ERA5 reanalysis dataset provides excellent ground-truth data.
>
> > Do the authors consider other models for baseline models?
>
> Please see our general comment; we have added additional baselines and discussion of our choices.
>
> References:
>
> [1] Grönquist et al., “Deep Learning for Post-Processing Ensemble Weather Forecasts”, _Philosophical Transactions of the Royal Society A_, 2021.
>
> [2] Gneiting et al., “Calibrated Probabilistic Forecasting Using Ensemble Model Output Statistics and Minimum CRPS Estimation”, _Monthly Weather Review_ 133(5), 2005.
>
> [3] Rasp & Lerch, “Neural Networks for Post-Processing Ensemble Weather Forecasts”, _Monthly Weather Review_ 146(11), 2018.
>
> [4] Li et al., “Convolutional Neural Network-based Statistical Post-Processing of Ensemble Precipitation Forecasts”, _Journal of Hydrology_ 605, 2022.

---

### Official Review · Reviewer_GFk8 · 2022-07-28

**Rating:** 9
**Confidence:** 2
**Clarity:** Yes.

**Strengths:**

- The data is composed of 20 years wether data extracted for 10 ensembles that is obtained for a weather simulator.
- The authors provide new ways of measuring the prediction correction task.
- The dataset includes important wether prediction parameters like height above sea level, pressure values, etc.
-

**Weaknesses:**

- Is the numerical weather prediction approach the most famous for weather prediction?
- Line 91, why did you use these measurements? Are there other parameters that you could have used?
- Line 173, similarly, why did you use these parameters?
- Why did you stop the data range at 2017 although we are in 2022?

**Additional Feedback:**

None.

**Correctness:**

Yes, the data is constructed by extracting weather data from a global weather simulator.

**Documentation:**

The documentation is clear.

**Ethics:**

No ethical concerns are related to the paper.

**Relation To Prior Work:**

Yes, although it would be nice to expand on the advantage of ensemble processing methods over deterministic data-driven forecasting.

**Summary And Contributions:**

This dataset paper aim to provide a post-processing prediction system data that can enhances the wether prediction. The dataset covers 20 years and has 10 ensemble members.

---

> ### Author Response · Authors · 2022-08-19
> **Initial Response**
>
> > Is the numerical weather prediction approach the most famous for weather prediction?
>
> The Integrated Forecast System, which generated the data we used to produce ENS-10, is the official model of the ECMWF and agreed upon by its 34 member and cooperating states. It is widely considered to be the most accurate global forecast model [1]. More generally, numerical weather prediction systems are by far the most widely used approach for weather forecasting.
>
> > Line 91, why did you use these measurements? Are there other parameters that you could have used?
>
> We believe you're referring to the description of the parameters used in the IFS model that generated ENS-10. This is the configuration used by the production model, and we will note this in the paper. This model and its configuration has been tuned for decades and it is not feasible to adjust it.
>
> > Line 173, similarly, why did you use these parameters?
>
> Please see our general comment. In short, to keep the dataset size reasonable, we selected key parameters based on community consensus and meteorological expertise.
>
> > Why did you stop the data range at 2017 although we are in 2022?
>
> Please see our general comment.
>
> **References:**
>
> [1] Haiden et al. “Evaluation of ECMWF Forecasts, Including the 2021 Upgrade”, _ECMWF Technical Memoranda_ 884, 2021. [http://dx.doi.org/10.21957/90pgicjk4]

---

> > ### Comment · Reviewer_GFk8 · 2022-08-24
> > **Clear explanation and detailed response**
> >
> > I thank the authors for their detailed and prompt responses to my questions.
> > I find the modifications and the answers clear and enhance the paper's readability. Hence, I am changing my score to 9.
> >
> > Again, thank you for your contribution and I wish you the best of luck.

---

### Author Response · Authors · 2022-08-19
**General Response**

We thank the reviewers for their comments and suggestions. We appreciate that reviewers found the dataset important, that it addresses a major gap for the community, and that it is well-documented and maintained.

We have updated the paper based on the feedback, and respond to common questions below and to specific reviewer questions in individual replies.

**Additional baseline models and how we selected the models:**

We have added two additional benchmarks to our evaluations: a LeNet-style convolutional model and a transformer-based model. Please see the updated table below (or in the updated paper):

**CRPS:**

| Model       | Z500 (5-ENS)     | Z500 (10-ENS)     | T850 (5-ENS)    | T850 (10-ENS)   | T2m (5-ENS)     | T2m (10-ENS)    |
|-------------| ---------------- | ----------------- | --------------- | --------------- | --------------- | --------------- |
| Raw         | 81.03            | 78.24             | 0.748           | 0.719           | 0.758           | 0.733           |
| EMOS        | 79.08±0.739      | 81.74±6.131       | 0.725±0.002     | 0.756±0.052     | 0.718±0.003     | 0.749±0.054     |
| MLP         | 75.84±0.016      | 74.63±0.029       | 0.701±2e-4      | 0.684±4e-4      | 0.684±6e-4      | 0.672±5e-4      |
| LeNet       | **75.56±0.101**  | **74.41±0.109**   | 0.689±2e-4      | 0.674±2e-4      | 0.669±7e-4      | 0.659±4e-4      |
| U-Net       | 76.66±0.470      | 76.25±0.106       | 0.687±0.003     | 0.669±0.009     | 0.659±0.005     | 0.644±0.006     |
| Transformer | 77.3±0.061       | 74.79±0.118       | **0.686±0.002** | **0.665±0.002** | **0.649±0.004** | **0.626±0.004** |

**EECRPS:**

| Model       | Z500 (5-ENS)    | Z500 (10-ENS)  | T850 (5-ENS)    | T850 (10-ENS)   | T2m (5-ENS)     | T2m (10-ENS)    |
|-------------| --------------- | -------------- | --------------- | --------------- | --------------- | --------------- |
| Raw         | 29.80           | 28.78          | 0.256           | 0.246           | 0.258           | 0.250           |
| EMOS        | 29.10±0.187     | 30.13±2.166    | 0.248±3e-4      | 0.259±0.018     | 0.245±0.001     | 0.255±0.018     |
| MLP         | 27.86±0.006     | 27.41±0.010    | 0.240±1e-4      | 0.234±2e-4      | 0.233±2e-4      | 0.229±2e-4      |
| LeNet       | **27.72±0.039** | **27.3±0.037** | 0.235±5e-5      | 0.230±8e-5      | 0.228±2e-4      | 0.224±1e-4      |
| U-Net       | 27.98±0.240     | 27.61±0.490    | 0.235±0.003     | 0.230±0.002     | 0.223±5e-4      | 0.219±0.001     |
| Transformer | 28.35±0.026     | 27.42±0.047    | **0.235±0.001** | **0.227±0.001** | **0.222±0.001** | **0.214±0.001** |

We have also clarified why we selected these models as baselines. While there are a number of works on ensemble post-processing, they broadly fall into five categories and our benchmark models serve as exemplars of them:
* “Simple” learned statistical methods, of which EMOS is a common example [1, 2, 3]. Note that these methods are also broadly representative of what is presently used by production forecast systems for post-processing.
* MLPs have been used by several different works [3, 4].
* Simple convolutional neural networks have also been used (e.g., by [5]), hence our choice of a LeNet-style CNN. Also, while it is a different task, CNNs have also been used by the community for weather prediction on WeatherBench [6, 7, 8].
* U-Nets have been used by [1], and are one of the more complex network architectures applied to post-processing.
* Finally, we have added results using a one-layer self-attentive ensemble transformer, which has recently been used for representing ensemble interactions in earth systems models [9]. Transformers have also recently been applied successfully to weather prediction [10].

---

> ### Author Response · Authors · 2022-08-19
> **General Response (part 2)**
>
> **How were the parameters used in ENS-10 selected:**
>
> This is a great question, and we have clarified this in the updated version of the paper.
>
> In brief, the production weather model that generated the data (ECMWF’s Integrated Forecast System) uses ten prognostic variables per grid point and 91 grid points per vertical column, plus additional prognostic variables at the surface. Since there are around one thousand prognostic variables per grid column, we need to reduce this number to keep the data volume reasonable. (The raw data would be hundreds of TB in size.)
>
> We therefore selected the majority of the variables (seven) on each of a subset (eleven) of the pressure levels (which are derived from model levels), and eleven surface variables. Unfortunately, there is no theory for which variables will work best, but we followed a combination of community consensus (e.g., [1, 3, 6, 9]) and expert meteorological advice from ECMWF in making our choice. Geopotential is quite meaningful as it provides information about the larger synoptic situation. Temperature and the U and V wind components are highly relevant for forecast users. Finally, specific humidity, vertical velocity, and divergence provide information about the cloud fields and convective state of the grid column.
>
> **Data beyond 2017:**
>
> We do not use data beyond 2017 for a couple of practical reasons. When the data to form ENS-10 were retrieved and processed from ECMWF archives, only data through 2017 were extracted. There are also downsides to producing a newer version of the hindcasts:
> * The version of the Integrated Forecast System that generates the hindcasts changes frequently (as the parameterization is tuned, the model improved, etc.). This means the properties of the meteorological fields would change as well. Thus, we cannot simply “append” to the dataset.
> * Second, hindcasts are run only for a certain number of years (currently twenty) back until the present. Thus, we would essentially be constructing a second dataset, albeit the first years of which would overlap with the last years of ENS-10.
>
> We clarify this in the paper.
>
> **References:**
>
> [1] Grönquist et al., “Deep Learning for Post-Processing Ensemble Weather Forecasts”, _Philosophical Transactions of the Royal Society A_, 2021.
>
> [2] Gneiting et al., “Calibrated Probabilistic Forecasting Using Ensemble Model Output Statistics and Minimum CRPS Estimation”, _Monthly Weather Review_ 133(5), 2005.
>
> [3] Rasp & Lerch, “Neural Networks for Post-Processing Ensemble Weather Forecasts”, _Monthly Weather Review_ 146(11), 2018.
>
> [4] Ghazvinian et al., “A Novel Hybrid Artificial Neural Network - Parametric Scheme for Postprocessing Medium-range Precipitation Forecasts”, _Advances in Water Resources_ 151, 2021.
>
> [5] Li et al., “Convolutional Neural Network-based Statistical Post-Processing of Ensemble Precipitation Forecasts”, _Journal of Hydrology_ 605, 2022.
>
> [6] Rasp et al., “WeatherBench: A Benchmark Data Set for Data-Driven Weather Forecasting”, _Journal of Advances in Modeling Earth Systems_ 12(11), 2020.
>
> [7] Weyn et al., “Improving Data-Driven Global Weather Prediction Using Deep Convolutional Neural Networks on a Cubed Sphere”, _Journal of Advances in Modeling Earth Systems_ 12(9), 2020.
>
> [8] Rasp et al., “Data-Driven Medium-Range Weather Prediction With a Resnet Pretrained on Climate Simulations: A New Model for WeatherBench”, _Journal of Advances in Modeling Earth Systems_ 13(2), 2021.
>
> [9] Finn, “Self-Attentive Ensemble Transformer: Representing Ensemble Interactions in Neural Networks for Earth System Models”, _Tackling Climate Change with Machine Learning workshop at ICML_, 2021.
>
> [10] Pathak et al., “FourCastNet: A Global Data-driven High-resolution Weather Model using Adaptive Fourier Neural Operators”, arXiv 2022.

---

### Meta-Review · Area_Chair_LMqq · 2022-09-09

**Recommendation:** Accept
**Confidence:** 4

**Metareview:**

This is a high quality, high impact dataset for weather forecasting, with a novel task and metrics to measure the quality of prediction by forecast ensembles. The reviewers are in agreement that this paper should be accepted. Congratulations.

---

### Decision · Program_Chairs · 2022-09-16

Accept